# Spatio-Temporal Variation of Gender-Specific Hypertension Risk: Evidence from China

**DOI:** 10.3390/ijerph16224545

**Published:** 2019-11-17

**Authors:** Li Xu, Qingshan Jiang, David R. Lairson

**Affiliations:** 1Department of Statistics, School of Mathematics and Statistics, Guangdong University of Foreign Studies, Guangzhou 510006, China; 201810006@oamail.gdufs.edu.cn; 2Division of Management Policy and Community Health, School of Public Health, University of Texas Health Science Center at Houston, 1200, Herman-Pressler Street, Houston, TX 77030, USA; david.r.lairson@uth.tmc.edu

**Keywords:** China, hypertension, spatio-temporal variation, Shared Component Model (SCM), Besag, York, and Mollie (BYM)

## Abstract

Previous studies which have shown the existence of gender disparities in hypertension risks often failed to take into account the participants’ spatial and temporal information. In this study, we explored the spatio-temporal variation for gender-specific hypertension risks in not only single-disease settings but also multiple-disease settings. From the longitudinal data of the China Health and Nutrition Survey (CHNS), 70,374 records of 21,006 individuals aged 12 years and over were selected for this study. Bayesian B-spline techniques along with the Besag, York, and Mollie (BYM) model and the Shared Component Model (SCM) model were then used to construct the spatio-temporal models. Our study found that the prevalence of hypertension in China increased from 11.7% to 34.5% during 1991 and 2015, with a higher rate in males than that in females. Moreover, hypertension was found mainly clustered in spatially adjacent regions, with a significant high-risk pattern in Eastern and Central China while a low-risk pattern in Western China, especially for males. The spatio-temporal variation of hypertension risks was associated with regional covariates, such as age, overweight, alcohol consumption, and smoking, with similar effects of age shared by both genders whereas gender-specific effects for other covariates. Thus, gender-specific hypertension prevention and control should be emphasized in the future in China, especially for the elderly population, overweight population, and females with a history of alcohol consumption and smoking who live in Eastern China and Central China.

## 1. Introduction

Hypertension is the leading cause of cardiovascular diseases in China. According to the China Guideline for the Prevention and Treatment of Hypertension, there are at least 1.5 million hypertension-associated deaths each year, accounting for more than half of the cardiovascular-associated deaths [1]. The October 2012 to December 2015 national survey reported that inhabitants in China aged 18 or older had a hypertension prevalence of 23.2% with a rate in males higher than that in females (24.5% vs. 21.9%), and large disparities between regions (ranging from 15.6% in Hunan to 35.9% in Beijing) were also observed [2]. This high prevalence of hypertension in China has placed a heavy burden of medical and social resource utilization on families and the whole society.

So far, research has identified risk factors for hypertension at the individual level [2,3,4,5,6,7]. However, less attention has been paid to gender-specific hypertension risk [8,9,10,11]. Moreover, traditional statistical methods used in hypertension research, such as logistic regression, Poisson regression, the Chi-Square test often failed to utilize the participants’ spatial or temporal information [2,3,4,5,6,7,8,9,10,11]. The spatial or spatio-temporal variation of hypertension, however, may provide valuable clues for identifying and quantifying risk factors for hypertension at regional levels, thus better informing guidelines for hypertension prevention and control at regional levels.

Among various epidemiological methods for investigating spatial or spatio-temporal variations for disease risks, disease mapping models have received increasing attention. By borrowing information across different geographical regions (or time windows), the pooled sample size in disease mapping studies will be large enough to obtain more reliable estimators of disease risks. Besides, with the advances of Markov Chain Monte Carlo (MCMC) approaches, disease mapping studies have become particularly popular due to its efficiency and flexibility. In addition, many outputs derived from the disease mapping models could be easily presented in the form of maps. These maps could provide a rapid summary of complicated geographic information visually, which may offer insight into the subtle patterns that are missed in tabular presentations [12].

This study was motivated by the fact that most disease mapping studies focused on rare diseases in spatial dimensions [13,14,15,16,17,18] although utilizing both spatial and temporal information can provide more insight into disease pathogenesis. What is more, previous hypertension studies were conducted at the individual level, which may have neglected the potential correlation between individuals in spatial or temporal dimensions. Therefore, in this study, we attempt to incorporate both spatial and temporal information into the spatial Besag, York, and Mollie (BYM) model and the Shared Component Model (SCM) to explore the variation and evolution of gender-specific hypertension risks in both single-disease and multiple-disease settings. By conducting the spatio-temporal analysis, the spatio-temporal variation of gender-specific hypertension risks can be better depicted, thus providing us important clues to investigate the potential gender-specific hypertension risk factors at regional levels.

This paper is organized as follows: Section 2 describes data, methods, and implementation issues. Section 3 presents the spatio-temporal modeling framework, and then specifies the formulas of the BYM and SCM models. The methodology is further illustrated in Section 3 with an empirical study for exploring the gender-specific hypertension risk in seven major provinces of China from 1991 to 2015. Discussions are in Section 4, and the conclusions are presented in Section 5.

## 2. Materials and Methods

### 2.1. Materials

#### 2.1.1. Data Source

Data were extracted from the China Health and Nutrition Survey (CHNS) by registering an account on the official website. CHNS is an ongoing large-scale and household-based survey in China, started in 1989 and the participants were followed up every 2–4 years. So far, there are a total of ten waves for this survey: 1989, 1991, 1993, 1997, 2000, 2004, 2006, 2009, 2011, and 2015. A stratified multistage sampling design was employed. All participants were provided with written informed consent, and the study was reviewed and approved by the University of North Carolina and the National Institute for Nutrition and Health (NINH) at the Chinese Center for Disease Control and Prevention (CCDC). Details of data collection method are available at official website: https://www.cpc.unc.edu/projects/china/about/design/datacoll. This study of CHNS was approved by the institutional review boards of the University of North Carolina at Chapel Hill and the National Institute for Nutrition and Health, Chinese Center for Disease Control and Prevention. Written Informed Consent was provided by each participant (07-1963).

As the first waves of survey, the sample of 1989 is small, and the surveyed regions were a little different. In order to obtain reliable results, the sample of 1989 were excluded in the spatio-temporal modeling. Therefore, the 1991 to 2015 CHNS survey data were utilized for analysis in this study. What is more, some records in the dataset were excluded according to our eligibility criteria: missing data of age, sex, systolic blood pressure (SBP) or diastolic blood pressure (DBP) (*n* = 9062); and age below 12 years (*n* = 16,194). Additionally, records of regions that were included only in one or two waves were excluded as well (*n* = 23,859) in order to capture the temporal pattern of hypertension risk. In the end, 70,374 records of 21,006 individuals aged 12 years and over were retained. Comparison was then made between the selected sample and the whole survey data (Table 1). It is clear that the selected sample is reasonably representative of the larger group.

#### 2.1.2. Study Areas

According to the selection criteria mentioned above, seven major provinces that were included in every survey were chosen in our study: Jiangsu, Shandong, Henan, Hubei, Hunan, Guangxi, and Guizhou. Furthermore, according to the geographical location, the surveyed provinces were categorized as Eastern China (Shandong, Jiangsu), Central China (Henan, Hubei, Hunan), and Western China (Guizhou, Guangxi) (Figure 1).

#### 2.1.3. Measurements and Definitions

Anthropometric data, including body weight, waist circumstances, and height were measured by standard protocols so that body mass index (BMI) could be calculated consistently. The overweight participants were people whose BMI was 24.0 kg/m^2^ or more [19].

Blood pressure was measured on the right arm by a physician, nurse, health worker, or other health professional using a standard mercury sphygmomanometer. In order to improve accuracy, it was taken three times, and their mean was then used in our analyses. Hypertension was defined as SBP ≥ 140 mmHg or DBP ≥ 90 mmHg or currently taking an antihypertensive drug based on the World Health Organization (WHO) criteria and defined.

### 2.2. Methods

In order to implement BYM and SCM modeling, Moran’s I, a correlation coefficient that measures the overall spatial autocorrelation was first calculated for the whole study period. The Bayesian B-Spline technique was then incorporated into the spatial BYM model to construct a spatio-temporal BYM model for a separate analysis. The Bayesian B-Spline technique was also incorporated into the spatial SCM model to construct a spatio-temporal SCM model for a joint analysis, and the results were then compared with those of the spatio-temporal BYM model. Finally, regional covariates, such as proportion of the elderly population (≥60), proportion of overweight population (BMI ≥ 24), proportion of ever smokers, and proportion of alcohol consumption were incorporated into the models to reveal the potential source of spatio-temporal variation.

Data extraction, data management, and basic statistical analyses were performed in STATA version 13.1 (StataCorp LLC, College Station, TX, USA). Maps of study areas were first produced in ArcMap version 10.3.1 (ESRI, Redlands, CA, USA), and then imported into GeoBUGS. All spatial adjacent weight matrices were created using GeoBUGS. B-Spline basis were computed with R version 3.5.0, and then employed during the spatio-temporal modeling. The spatio-temporal BYM and SCM models were implemented in OpenBUGS version 3.2.3.

#### 2.2.1. Rationale and Motivation of Spatio-Temporal Models

The BYM model, introduced by Besag, York, and Mollie [20], is widely used for single disease mapping [21,22,23,24,25,26,27,28], whereas the SCM, introduced by Knorr and Best and extended by Held et al. [13,29], is usually applied in a multiple disease settings [14,15,16,17,18,30,31,32]. Most previous studies of BYM model and SCM model focused on rare diseases in spatial dimension. With the availability of spatio-temporal data in recent decades, the spatio-temporal models have attracted more and more attention. There are two major concerns for spatio-temporal modeling. The first is whether the interaction effect is significant, and the second is how to fit temporal effects.

For the first major concern, there were mainly three opinions. Some researchers claimed that the inclusion of the interaction effect would make a much more complex model and induce identifiability problems [21]. Others argued that the heterogeneity and interaction terms are competing to explain the space-time structure not captured by the main effects [15,16]. Some then suggested considering interaction effects for a chronic disease in a long period, usually over 15 years [15,33].

For the second major concern, solutions depend on whether the temporal effects are fitted globally or locally. For global model fitting, a linearity and an auto-regression were a most popular assumption [15,34,35,36]. For local model fitting, splines were found appealing in spatio-temporal modeling in disease mapping studies owing to its flexibility, especially when data are available for several time periods (e.g., nine or more) [27,37,38].

Though spatio-temporal Bayesian disease mapping models may be a good way to capture the spatio-temporal effect [15,16,33,34,35,36,37,38,39,40,41,42,43,44], the use of the spatio-temporal BYM model or the SCM models is not so common in the literature [15,16,36,42,43]. Additionally, previous studies mainly focused on rare diseases in a short time period (e.g., five time periods with the same interval) [14,16]. Besides, in some studies, data were aggregated over several years due to the observation sparsity [13,15]. It is obvious that hypertension is not a rare disease. Therefore, some modifications are needed with regard to the model formulation. In this study, the data feature was different from most previous studies in that it covered a long period with unequal intervals between surveys. Under such circumstances, we incorporated Bayesian B-spline into the spatial BYM model and the SCM model; and constructed spatio-temporal Bayesian disease mapping models to accommodate the spatio-temporal variation of gender-specific hypertension risk in single disease and multiple disease settings, respectively.

#### 2.2.2. Spatio-Temporal BYM Model

The spatio-temporal BYM model was constructed as follows:(1)$$O_{it}\sim bin\left(n_{it},p_{it}\right),$$
(2)$$logit\left(p_{it}\right)=\alpha_{0}+S_{0}\left(t\right)+b_{i0}+RS_{i}\left(t\right),$$
(3)$$b_{i0}=u_{i0}+s_{i0},$$
where *O_it_* and *n_it_* represent the observed and the total number of cases of hypertension for region *i* (*i* = 1, …, N) and time period *t*(*t* = 1, …, T), respectively, and *p_it_* denotes the prevalence of hypertension for region *i* and period *t*. $u_{i0}$ and $s_{i0}$ represent the unstructured and spatially structured random-effect, respectively, and $\alpha_{0}$ represents the fixed-effect. Separate models were constructed for males and females. In order to accommodate different time intervals between consecutive surveys, the year t was centered at the year 2004 [39,42]. Both $S_{0}\left(t\right)$ and $RS_{i}\left(t\right)\;$ are smooth functions of spline, representing the smoothed global and local time trends, respectively. $\alpha_{0}+S_{0}\left(t\right)$ and $b_{i0}+RS_{i}\left(t\right)$ represent the time trends of global and local disease risk, respectively.

So far, previous disease mapping models mainly focused on rare diseases; thus, Poisson distribution was assumed with the log link function. However, hypertension is not a rare disease and the binomial distribution assumption is more appropriate, so the logit link function is employed in our study [30].

Moreover, even though both smoothing splines and P-splines are commonly used methods, they only allow for limited smoothing prior options. Additionally, due to the large number of unknown parameters, models with smoothing splines and P-splines may not work well when data are limited, which is typically the case in disease mapping studies. Therefore, these methods may suffer from considerable uncertainty if the sample size is not sufficiently large [40]. In contrast, the regression B-spline allows for more smoothing prior options and has its analytic and computational advantages [41]. Thus, regression B-spline is employed in our study to fit the temporal trend.

For the cubic B-spline basis function with L knots, $S_{0}\left(t\right)$ and $RS_{i}\left(t\right)$ can be rewritten as follows:
(4)$$S_{0}\left(t\right)=\overset{K}{\underset{k=1}{\sum }}a_{k}B_{k}\left(t\right),$$

(5)$$RS_{i}\left(t\right)=\overset{K}{\underset{k=1}{\sum }}b_{ik}B_{k}\left(t\right),$$
where $B_{k}$ (*k* = 1, …*K*) is a set of B-spline basis functions, $B_{k}\left(t\right)$ denotes the k-th B-spline basis function evaluated at time *t* (*K* = *L* + 3), $a_{k}$(*k* = 1, …, *K*) are the fixed-effect spline coefficients, and *b*_*k*_(*b*_1*k*_, …, *b_nk_*)*^T^* are the random-effect spline coefficients. However, to ensure the identifiability of the model, the intercept term is not included in this study.

At the same time, the number of knots is crucial: a small number of knots may not be sufficient to capture the variability of the data, yet a large number of knots may lead to overfitting [44]. The position of the knots is also important, especially when there are not enough knots [44,45]. As suggested by Macnab and Gustafson (2007), the ‘forward selection’ approach should be applied for selection of knots [40]. We then adopted this approach with some modifications due to the unequally spaced time intervals of our data. Specifically, one knot was initially set at the year of 2004 (the center) rather than the mean *t*. The other knots were set according to the results of the descriptive analysis. Overall the number of knots increased gradually, and they were evaluated based on the deviance information criterion (DIC).

##### Prior Distributions

It is well known that in Bayesian inferences, all unknown parameters, including random spline coefficients, are random variables which require assignment of prior distributions. Thus, in this study, all prior distributions for parameters and hyper parameters were assigned based on previous studies [16,18] and the structure of our dataset.

Similar to the spatial BYM model, a convolutional prior is adopted for $b_{i0}$, and $u_{i0}$ and $s_{i0}$ were assumed to follow a Conditional Autoregressive Regression (CAR) prior and a Gaussian prior (with precision parameters $\tau_{u}$ and $\tau_{s}$), respectively. Fixed-effect parameters ($\alpha_{0}$ and $\alpha_{k}$) were then assumed to have a non-informative prior. The random spline coefficients $b_{ks}$*s* were assigned to the Multivariate Conditional Autoregressive Regression (MCAR) with condition mean and precision matrix as follows:$$\pi \left(b\right)\sim MVN\left(0,{Ω}^{-1}\right),$$
Ω=(L−W)⊗Γ,
where $W=(w_{ij})$ is a spatial adjacent matrix, and *L* is a *K* by *K* identity matrix, Γ
and Ω are a *K* by *K* and a *NK* by *NK* precision matrix, respectively, with Γ being symmetric positive definite.

All priors for fixed-effect parameters were assumed to follow the N (0, 10,000) distribution. The priors for precision parameters $\tau_{u}$ and $\tau_{s}$ were then assumed to follow a gamma (5, 0.0005) distribution. The precision matrix Ω was assumed to follow the Wishart ($R^{-1},p$) distribution, and the hyper parameters R and p (p≧K) were set as identity matrix and *K*, respectively. Additionally, *p* was set equal to *K* to represent a non-informative prior.

#### 2.2.3. Spatio-Temporal SCM Model

The spatio-temporal SCM model was constructed as follows:(6)$$O_{jit}\sim bin\left(n_{jit},p_{jit}\right),$$
(7)$$logit\left(p_{jit}\right)=\alpha_{j}+eta_{jit},$$
(8)$$eta_{1it}=\left(b_{0i}+RS_{0i}\left(t\right)\right)*\delta_{t}+S_{1}\left(t\right)+b_{1i}+RS_{1i}\left(t\right)+\beta_{it},$$
(9)$$eta_{2it}=\left(b_{0i}+RS_{0i}\left(t\right)\right)/\delta_{t}+S_{2}\left(t\right)+b_{2i}+RS_{2i}\left(t\right)+\beta_{it},$$
(10)$$\eta_{1}\left(t\right)=var\left(\left(b_{0i}+RS_{0i}\left(t\right)\right)*\delta_{t}\right)/var\left(eta_{1it}\right),$$
(11)$$\eta_{2}\left(t\right)=var\left(\left(b_{0i}+RS_{0i}\left(t\right)\right)/\delta_{t}\right)/var\left(eta_{2it}\right),$$
where *α_j_* is the gender-specific intercept (*j* = 1 for males and *j* = 2 for females), representing the gender-specific baseline hypertension risk. *b_0i_* and *RS_0i_*(*t*) denote the shared spatial and spatio-temporal variations for both genders, respectively; therefore, (*b_0i_ + RS_0i_*(*t*)) is the overall shared variation for males and females. *S_j_*(*t*) and *b_ji_* represent the gender-specific temporal and spatial variations, respectively, and *RS_ji_*(*t*) represents the gender-specific spatio-temporal variation. *β_it_* is the spatio-temporal variation. *η_j_*(*t*) represents the proportion of variance explained by the shared components. *δ_t_* and 1/*δ_t_* are weights of the shared components (*b_0i_ + RS_0i_*(*t*)) for males and females, respectively. The weighting method mentioned above (*δ_t_* and 1/*δ_t_*) set the sum of the logarithm weight equal to 0 in order to ensure identifiability of the model. The meanings of *O_jit_*, *n_jit_*, and *p_jit_* are similar to those of the spatio-temporal BYM model in Section 2.2.2 except that the parameters are three dimensional rather than two dimensional.

Here it is also worthwhile noting that *b_0i_* and *RS_0i_*(*t*) represent the shared spatial main effect and the shared spatio-temporal main effect, respectively; *S_j_*(*t*), *RS_ji_*(*t*), and *β_it_* represent the gender-temporal interaction effect, gender-spatio-temporal interaction effect and spatio-temporal interaction effect, respectively.

Similar to Equations (4) and (5), the smooth function *RS_0i_(t)*, *S_j_(t)* and *RS_ji_(t)* can be decomposed as follows:(12)$$RS_{0i}\left(t\right)=\overset{K}{\underset{k=1}{\sum }}b_{ik}B_{k}\left(t\right),$$
(13)$$S_{j}\left(t\right)=\overset{K}{\underset{k=1}{\sum }}a_{jk}B_{k}\left(t\right),$$
(14)$$RS_{ji}\left(t\right)=\overset{K}{\underset{k=1}{\sum }}\beta_{jik}B_{k}\left(t\right),$$
where *a_jk_* is the coefficient of the *k*-th fixed spline for gender *j* (*j* = 1, 2; *k* = 1, …*K*), and β*_jk_s* is the coefficient vectors of the kth random spline for gender *j*. Definitions of *b_ik_* and *B_k_* are similar to those of the spatio-temporal BYM model in Section 2.2.2. Additionally, the method of knots selection is similar to that of the spatio-temporal BYM model.

##### Prior Distributions

In order to incorporate the spatial structure, both *β_it_* and *β_jk_* were assumed to have a CAR prior with the corresponding precision hyper parameters *τ_β_* and *τ_b_* being assigned to have the gamma (5, 0.0005) prior. The logarithmic of weight *δ_t_* was assumed to have the N (0.0, 0.169) distribution to ensure that *δ*^2^ falls between 1/5 and 5 with a 95% probability [18]. Similar to the spatio-temporal BYM model in Section 2.2.2, the prior for the random-effects spline coefficients $b_{k}$ is MCAR, with the precision matrix *Γ_b_* assigned from Wishart ($R^{-1},p$), and the fixed-effects spline coefficients *a_jk_* should follow the distribution N (0, 10,000). Additionally, in order to improve the identifiability and reduce the model complexity, both *b_ji_* and *b_0i_* were also assumed to follow the distribution N (0, 10,000).

## 3. Results

### 3.1. Statistical Descriptions

In total, 70,374 records of 21,006 individuals aged 12 years and over were included in our study, of which 47.5% were males. From 1991 to 2015, the prevalence of hypertension in China increased from 11.7% to 34.5%, with a higher rate in males than that in females (increased from 12.8% to 38.1% vs. 10.8% to 31.5%, respectively).

Figure 2 shows the trend of overall and gender-specific hypertension prevalence. An upward trend was observed for both genders, especially after 2006. Additionally, gender variation of hypertension existed with males experiencing a higher rate than that in females, showing the necessity of gender-specific modeling.

Figure 3 and Figure 4 show the area-specific hypertension prevalence for males and females, respectively. Hypertension prevalence increased for almost all regions for both males and females, and the highest value was found in Shandong province before 2004 whereas it was replaced by Jiangsu province after that. However, geographic disparities existed for both genders. In particular, hypertension prevalence for Henan province increased rapidly after 2006 and it was ranked No.2 for males and No.1 for females in 2015. In contrast, hypertension prevalence in males and females in Guizhou province was increasing steadily before 2009 with a decline after that.

Furthermore, the global Moran’s I and corresponding *p*-value were calculated for all study periods in order to assess the spatial correlation of hypertension prevalence (Table 2). It showed that though fluctuated during the study periods, Moran’s I was significantly positive, with an exception for males in 1991. This indicated that the hypertension risk was significantly spatially clustered, thus, it is reasonable to construct a spatio-temporal model.

### 3.2. Model Selection

The Bayesian Spatio-Temporal BYM and SCM models were fitted using Markov Chain Monte Carlo (MCMC) techniques. To ensure reliability, two independent parallel MCMC chains were run for each model with a total of 100,000 iterations, keeping every 10th, after a 50,000 iteration burn-in period for each chain. The results were based on thinned sample sizes of 50,000. Brooks–Gelman–Rubin diagnostics [46], as well as graphical checks of chains and their autocorrelations were performed to assess convergence. The model selection was guided by the commonly used deviance information criterion (DIC) [47] after convergence. In other words, Dbar (model fitting) and pD (model complexity) were key considerations during model selection. For brevity, the detailed information about model selection is provided in Appendix B.

### 3.3. Results of the Selected Models

Based on the comparisons shown in Table A3 in Appendix B, D2 and D1F were the best model for males and females, respectively for the spatio-temporal BYM model; while E2S was preferred for the spatio-temporal SCM model. Overall, both the spatio-temporal BYM model and the SCM model were able to identify the spatio-temporal pattern of hypertension risks. In terms of model fitting and model complexity, the spatio-temporal BYM model was favored with a clearer spatial pattern. However, due to the necessity of modeling males and females separately, it takes twice as much as time to run the spatio-temporal BYM model (Table A3 in Appendix B), and the spatio-temporal SCM model could produce more outputs, such as the gender-specific and shared spatial variations, gender-specific temporal effects, and the weight of shared component for both genders.

Next, the major results derived from the spatio-temporal BYM model and the spatio-temporal SCM model are presented respectively, as follows:

#### 3.3.1. Spatio-Temporal Pattern of Hypertension Risk

Results derived from the spatio-temporal BYM model show that the overall hypertension risk increased over time for both males and females (Figure 5). However, gender variation existed in terms of the spatial pattern of hypertension risks: for males, geographical disparities emerged in 1997, with a relatively higher hypertension risk first found in Shandong province, then the hypertension risks started to cluster in Eastern China (Jiangsu and Shandong) and Central China (Henan, Hubei and Hunan) over time; for females, geographical disparities had been found since 2000 with a smaller magnitude of hypertension risk and a slower change of its spatial pattern. What is more, for males, it is worth mentioning that the hypertension risk was still at a medium size (0.5 < Posterior mean < 0.8) for most regions before 2011 even though the hypertension risk in Jiangsu province increased dramatically from 0.64 in 2011 to 1.06 in 2015. Thus, in the future, special attention regarding hypertension prevention and control should be given to Eastern China, especially to males in Jiangsu province.

For the overall spatio-temporal pattern of the hypertension risks derived from the spatio-temporal SCM model, it was similar to that derived from the BYM model for both males and females (Figure 6). However, a less clear spatial pattern was observed due to the smoothing effect between males and females. Besides, there has been a smaller magnitude of hypertension risk for males in Eastern China (Shandong and Jiangsu) and in most parts of Central China (Henan and Hubei) since 2009.

#### 3.3.2. Shared Spatial Effects, Gender-Specific Spatial Effects, and Gender-Specific Temporal Effects

In terms of the spatial effects derived from the spatio-temporal SCM, it suggested that the gender-specific variation played a major part in the overall spatial variation of the hypertension risk with a clear spatial pattern. In particular, a higher hypertension risk was observed in Jiangsu, Shandong, and Henan for males and Jiangsu for females (Figure 7a,b). In contrast, the overall shared spatial variation (Figure 7c) was at a medium size (Posterior mean < 0.8) with no significant spatial pattern. However, an upward trend was observed for the weight (δ) of the overall shared spatial variation over time, which indicated that the common hypertension risk factors played a more and more important part for males and females, even though the contribution of the shared hypertension risk factors was no more than 30% before 2009 (Figure 8).

In terms of the temporal effects derived from the spatio-temporal SCM, there existed a declining trend of the hypertension risk between 1991 and 1993 because there was negative gender-specific temporal effect during this period for both males and females. However, after 1993, an upward trend was observed for males with an abrupt increase in 1997, yet the increase became steady after 2000, and no significant effects was observed for females after 1993 (Figure 9). Thus, gender-specific hypertension prevention and control should be considered in the future, and special attention should be paid to males if no effects of regional covariates are taken into account.

#### 3.3.3. Spatio-Temporal Interaction Effects

The spatio-temporal interaction effects derived from the spatio-temporal BYM model and the spatio-temporal SCM model are displayed in Appendix A and Appendix A, respectively. It was further demonstrated that there were significant spatio-temporal interaction effects. However, different spatial patterns were captured in these two models: for the spatio-temporal BYM model, a clearer pattern of Eastern China, Central China, and Western China was identified, and the pattern changed frequently; whereas the spatial-temporal interaction effects derived from the spatio-temporal SCM model were found mainly clustered in Western China and Central China, and then gradually moved toward Eastern China with a larger value than that derived from the spatio-temporal BYM model.

#### 3.3.4. Effects of the Regional Covariates

Moreover, the effects of all the covariates including age, overweight, alcohol consumption, and smoking increased over time, especially for females having a history of drinking and smoking (Appendix A). Specifically, the regional hypertension risk in 2015 increased about 4% for participants aged 60 and over, overweight population, and males having a history of smoking (Appendix A). What is more, though no significant risk was found for males having a history of alcohol consumption, the regional hypertension risk of alcohol consumption for females in 2015 increased 6.5%. Furthermore, the regional hypertension risk in 2015 increased 44% for females having a history of smoking (Appendix A). Overall, similar effects of age on regional hypertension risks were found between males and females. Nevertheless, the effects of overweight, alcohol consumption, and smoking were gender-specific: there was a slightly larger impact on males than on females for overweight, yet there was much more significant impact on females than on males regarding alcohol consumption and smoking (Appendix A). However, the different impact of overweight on hypertension risks between males and females became smaller over time. These suggest that special attention might paid to the elderly population, overweight population, and females with a history of alcohol consumption and smoking.

To conclude, both the spatio-temporal BYM and SCM models are able to identify the spatial pattern and the evolution of the hypertension risk over time, with the former producing clearer spatial patterns. However, being able to model males and females simultaneously, the spatio-temporal SCM model is more efficient, and it is capable of revealing the gender-specific and shared spatial variations and gender-specific temporal effects. Thus, the spatio-temporal SCM model is more favorable when there are significant shared spatial effects. However, be aware that there may be a risk of over smoothing for the spatio-temporal SCM when no significant shared effects are found. Therefore, in terms of hypertension prevention and control, special attention should be given to the people living in Eastern China and Central China, especially for the elderly population, overweight population, and females with a history of alcohol consumption and smoking.

### 3.4. Convergence of Key Model Parameters and Sensitivity of Selected Models

To ensure the reliability of the estimation, convergence of model parameters was diagnosed, and the sensitivity of the selected model to the choice of priors for the precision parameters were also assessed. Overall, there was no obvious autocorrelations and convergence of key parameters were achieved (Figure 10 and Figure 11). The sensitivity analysis confirmed that the posterior estimates and the resulting DIC scores were both robust in regard to the moderate changes in the prior distributions, thus their appropriateness in the proposed models (Table 3).

For brevity, Figure 10 only presents the autocorrelation results for *b_1_*, *b_2_*, *S*, and *β* when i = 2, the results of other parameters can be obtained upon request. Note that S1 [2] means the gender-specific temporal effects in 1993 (Year No.2) and beta1 [2] means spatio-temporal interaction effects for area No.2 in year No.1 (that means Hunan province in 1991).

For brevity, Figure 11 only presents the convergence results for *b_1_*, *b_2_*, *S*, and *β* when i = 2, the results of other parameters can be obtained upon request. Note that S1 [2] means the gender-specific temporal effects in 1993 (Year No.2) and beta1 [2] means spatio-temporal interaction effects for area No.2 in year No.1 (that means Hunan province in 1991).

## 4. Discussion

High prevalence of hypertension in China has placed a heavy burden on families and the whole society. Though a lot of research has been done to identify risk factors for hypertension, less attention was paid to the gender-specific hypertension risk. In addition, traditional methods were often employed, and the data were usually analyzed at the individual level, which may have neglected the potential correlation between individuals in spatial or temporal dimension. In this paper, to address this issue, new spatial-temporal BYM and SCM models were constructed with the Bayesian B-spline technique. The methodology was further illustrated by exploring spatial distribution and the evolution of gender-specific hypertension risks in seven major provinces of China. In terms of model fitting, the spatio-temporal BYM model was more favorable. However, the spatio-temporal SCM model took much less time to run, and it was able to produce richer output for gender-specific hypertension prevention and control, such as the gender-specific and shared spatial variation, the weights of shared component for both genders, and gender-specific temporal effects.

Similar spatio-temporal patterns of hypertension risks were identified from the spatio-temporal BYM model and the spatio-temporal SCM model, with a clearer spatial pattern and larger magnitude derived from the former. Results showed that the hypertension risks were increasing over time for both males and females, with a higher rate in males than that in females. What is more, hypertension was mainly clustered in spatially adjacent regions, with a significant high-risk pattern in Eastern and Central China whereas a low-risk pattern in Western China. In particular, the hypertension risk for males in Jiangsu province has started to become salient since 2009. Thus, special attention should be paid to these areas for hypertension prevention and control, especially for males in Jiangsu province.

Although the effects of age, overweight, alcohol consumption, and tobacco use on hypertension are well documented at the individual level [2,3,4,5,6,7,8,9,10,11,48], their effects on hypertension has not been investigated yet at regional levels. Our study found that the spatio-temporal variation of hypertension risk was also associated with age, overweight, alcohol consumption, and tobacco use at regional levels. Similar effect of age between males and females was identified, and the effect of tobacco use on females was more significant than that on males while the effect of overweight on males was more significant, which was consistent with the findings of previous studies [3,8]. Besides, our study also identified a more significant impact of tobacco use on females, consistent with the findings of previous studies in China [10]. Though consistent with the finding of Hui et al. (2018) [10] that there was more impact of alcohol consumption on females, other researchers identified a more significant association between alcohol intake and hypertension risk for males [9,11,49]. This inconsistency may be due to the fact that only the status of ever drinking was considered in our study. However, the type-of alcohol, frequency, and the amount of alcohol intake may also matter [9,49].

Furthermore, the relatively higher hypertension risk for people living in spatially adjacent provinces (e.g., Jiangsu, Shandong, and Henan) may also be associated with the similar flavor preference in these areas, e.g., their higher salt diet [50,51,52]. Therefore, comprehensive strategies are urgently needed to focus on reducing salt intake, in addition to maintaining optimal body weight, abstaining from smoking and alcohol consumption in females, with special attention on the elderly population.

It is also worthwhile mentioning that the effects of all the covariates analyzed in our study increased over time, especially for females with a history of alcohol consumption and tobacco use. A significant increase of regional hypertension risks for females was not observed until 2009 for females with a history of alcohol consumption and tobacco use. Considering the significant accumulation effects of risk factors for females, emphasis should be put on females with a history of alcohol consumption and tobacco use regarding hypertension prevention and control.

The spatio-temporal SCM model further quantified the weight of shared hypertension risk factors between males and females at regional levels. We found that though it increased steadily, the effect of shared hypertension risk factors for males and females accounted for no more than 30% before 2009. This indicated that the common factors only contributed a small proportion, and it further verified the existence of gender-specific hypertension risks [8,10]. However, the weight increased significantly after 2009. This significant increase was associated with the decreasing difference between the effect of overweight for males and females. Thus, we may conclude that the effect of age was mainly shared before 2009 whereas both the effects of age and overweight were shared after 2009 for males and females. This also suggests that attention should be given to the time varying effect of hypertension risk factors in order to better guide the gender-specific hypertension prevention and control.

Our study also shows that there was a significant spatio-temporal interaction effect for hypertension. This supports previous results that an examination of spatio-temporal interaction effect is necessary for chronic diseases spanning over a long time period (e.g., 20 years) [15,33]. Besides, the spatio-temporal SCM model identified the gender-specific temporal trend of hypertension risks, in which males experienced a significant upward trend after 1993. This result further verified the existence of gender variation of hypertension risks and the necessity of gender-specific spatio-temporal modeling.

In short, by incorporating the Bayesian B-spline into the previous BYM and SCM models, the newly constructed models were able to explore and highlight the spatial pattern of hypertension risks over time, thus providing information for disease etiology study. The spatio-temporal framework illustrated in our study is also applicable for rare diseases if modifications are made to model formulation.

Of course, this study had some limitations. First, due to the availability of our data source, only seven major provinces of China were selected for the spatio-temporal modeling, so our study may not capture the spatio-temporal variation of gender-specific hypertension risks in other provinces in China. Second, our data were only analyzed at the province level, which may have led to considerable uncertainty for statistical inference. However, data at a finer scale (e.g., city or county) were not available due to the confidential policy related to the survey data. Third, no clear relationship between alcohol consumption and hypertension risk for males was identified in our study. This may be due to the fact that only the status of current alcohol use was taken into account. Fourth, the association between the excessive sodium intake and the higher hypertension risk in Jiangsu, Shandong, and Henan was not further analyzed due to time constraint. Therefore, future research should take into account more detailed information about alcohol consumption and sodium intake and further identify the potential gender-specific risk factors at regional levels.

## 5. Conclusions

The extended spatio-temporal models showed that the hypertension risk was increasing over time in China, and it is often clustered in spatially adjacent regions for both males and females. However, a clearer spatial pattern was observed for males, and there was a larger magnitude of hypertension risk for males than females. The similar pattern of hypertension risks between males and females was mainly associated with age before 2009 whereas both the effects of age and overweight were shared after 2009. Furthermore, the gender-specific pattern was mainly associated with overweight, alcohol consumption, and smoking. Thus, gender-specific hypertension prevention and control should be emphasized in the future in China, especially for the elderly population, overweight population, and females with a history of alcohol consumption and smoking living in Eastern China and Central China.

## Figures and Tables

**Figure 1 ijerph-16-04545-f001:**
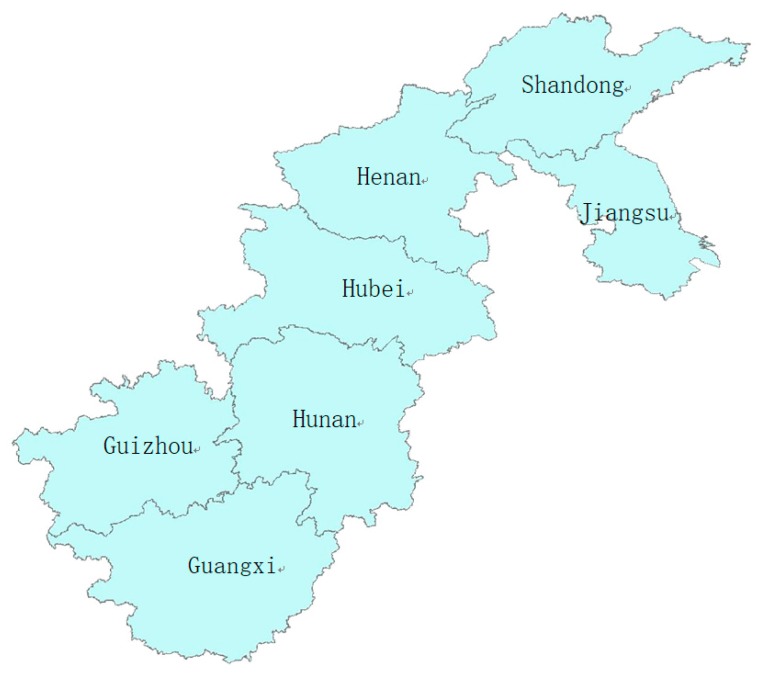
The Administrative Map of Study Areas.

**Figure 2 ijerph-16-04545-f002:**
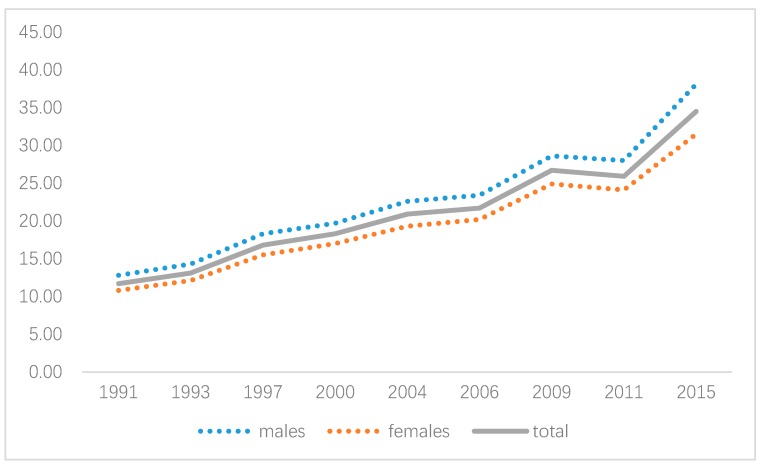
Hypertension prevalence during 1991 to 2015.

**Figure 3 ijerph-16-04545-f003:**
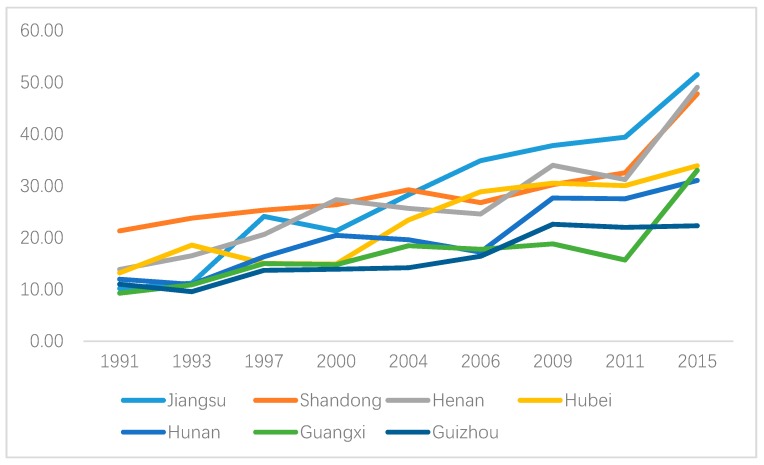
Area-specific hypertension prevalence (male).

**Figure 4 ijerph-16-04545-f004:**
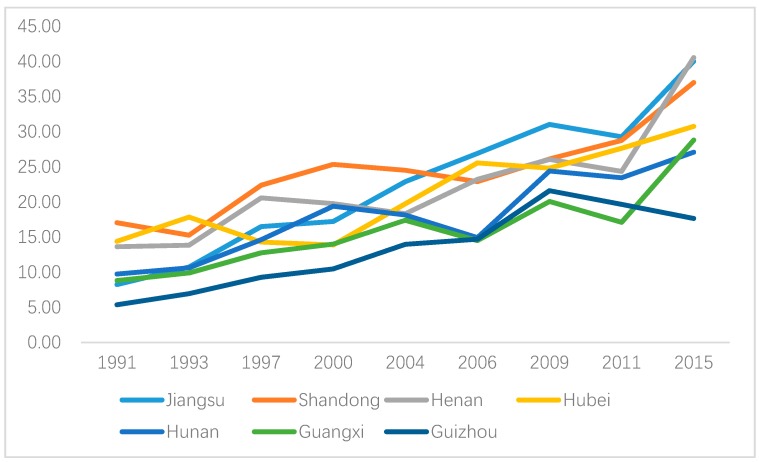
Area-specific hypertension prevalence (female).

**Figure 5 ijerph-16-04545-f005:**
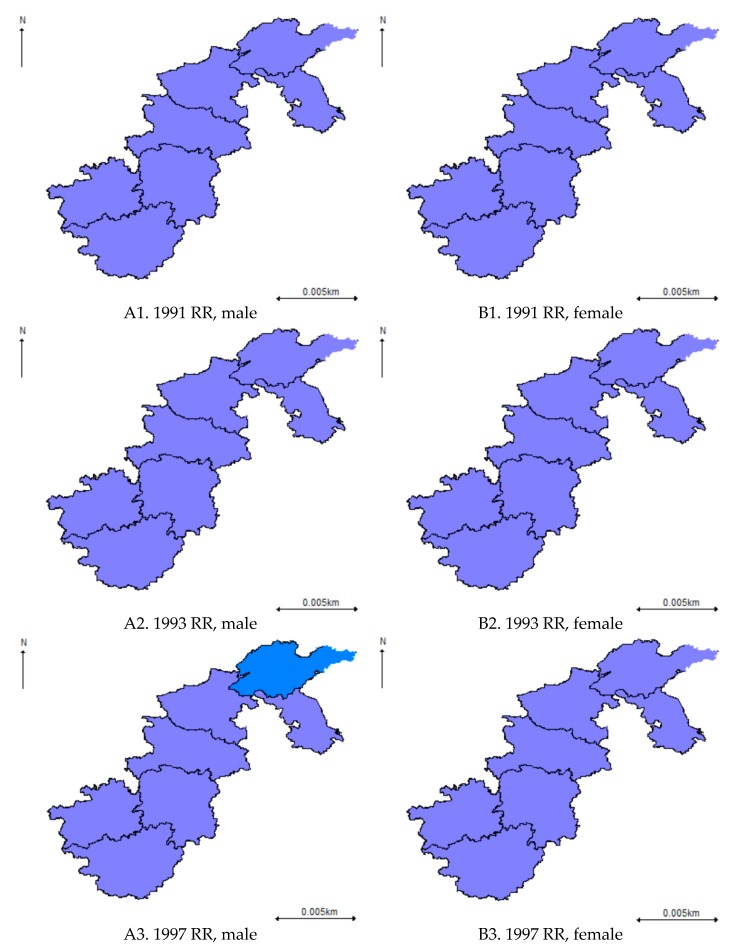
Posterior means of hypertension risk derived from the spatio-temporal Besag, York, and Mollie (BYM) model.

**Figure 6 ijerph-16-04545-f006:**
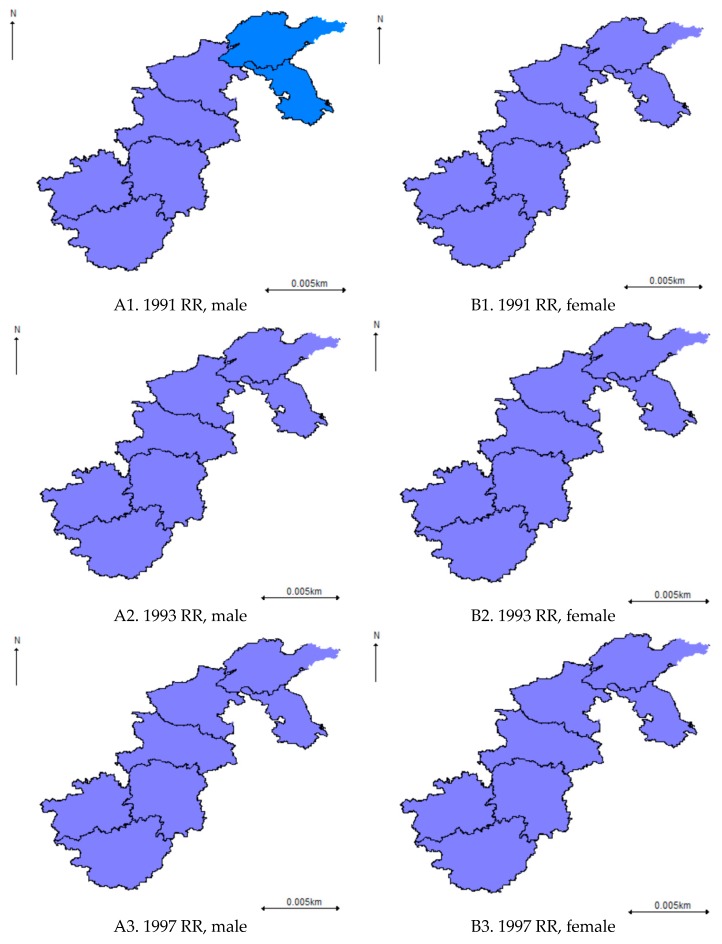
Posterior means of hypertension risk derived from the spatio-temporal Shared Component Model (SCM) model.

**Figure 7 ijerph-16-04545-f007:**
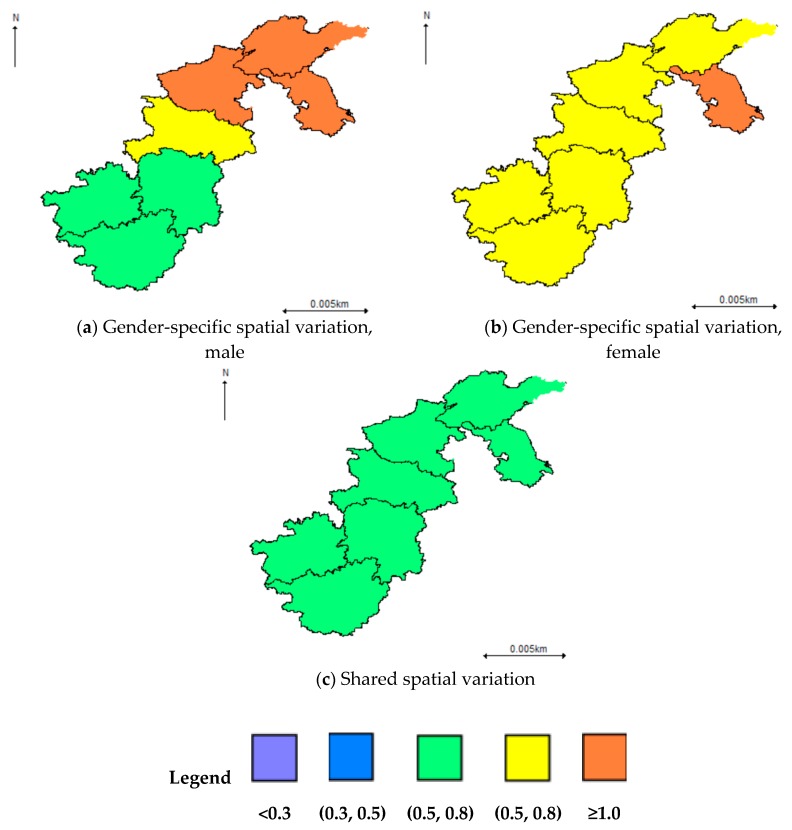
Overall shared and gender-specific variation derived from the spatio-temporal SCM model.

**Figure 8 ijerph-16-04545-f008:**
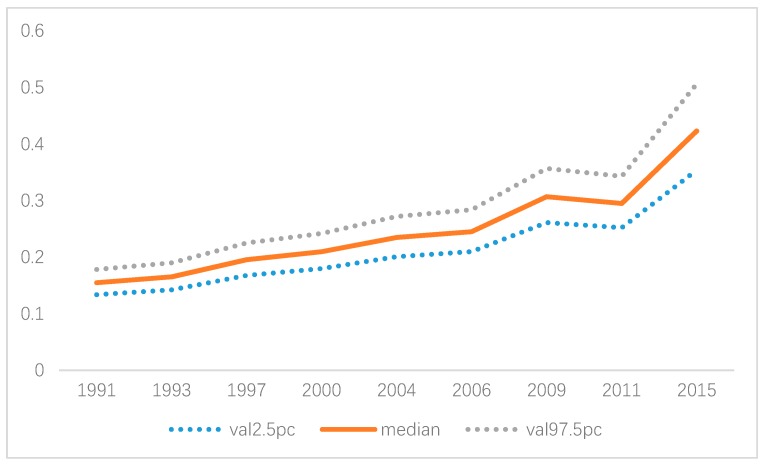
Weight parameter δ over time (posterior median and 95% CI).

**Figure 9 ijerph-16-04545-f009:**
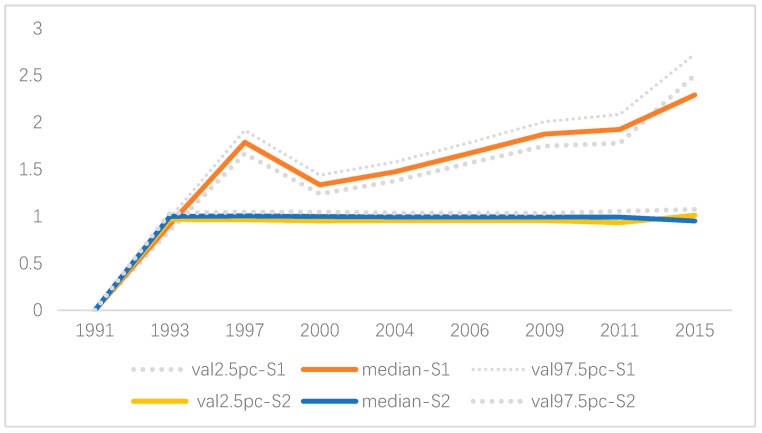
Gender-specific temporal effects (*j* = 1 for male; *j* = 2 for female).

**Figure 10 ijerph-16-04545-f010:**
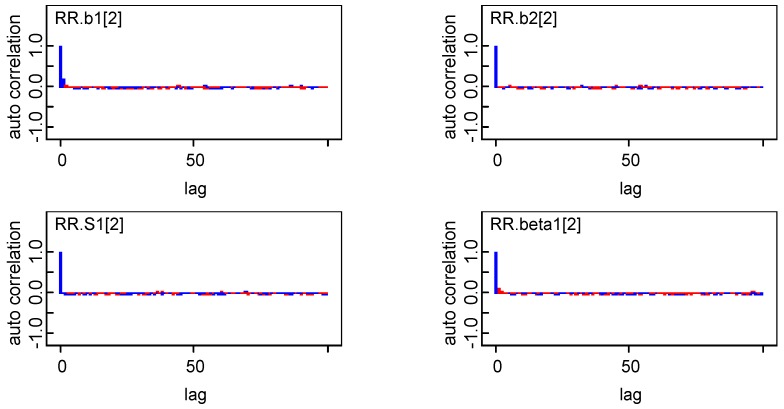
Autocorrelation of *b_1_*, *b_2_*, *S*, and *β*.

**Figure 11 ijerph-16-04545-f011:**
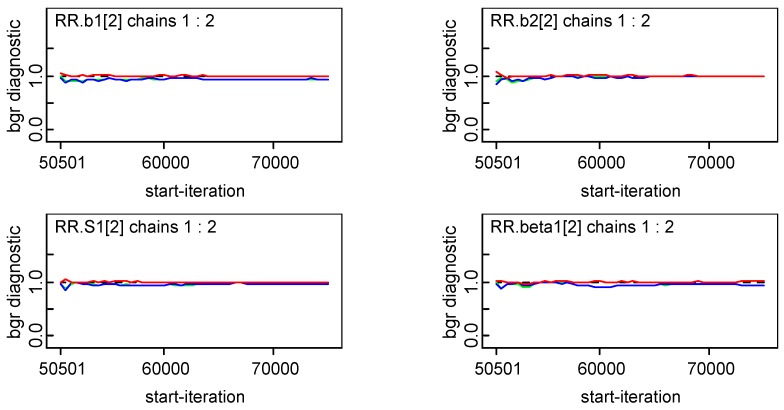
Convergence of *b_1_*, *b_2_*, *S*, and *β.*

**Table 1 ijerph-16-04545-t001:** Sample distribution of participants.

Variable	Value	Unit	Full Survey	Selected Sample
Gender	Male Female	%	49.0	47.5
		%	51.0	52.5
Urban	Rural Urban	%	66.4	67.5
		%	33.6	32.5

**Table 2 ijerph-16-04545-t002:** Global cluster for hypertension prevalence by gender.

Year	Moran’s *I* (Male)	*p*-Value (Male)	Moran’s *I* (Female)	*p*-Value (Female)
1991	−0.159	0.466	0.166	0.161
1993	0.074	0.204	0.273 *	0.091
1997	0.763 ***	0.003	0.508 **	0.022
2000	0.258	0.143	0.072	0.242
2004	0.775 ***	0.002	0.526 **	0.011
2006	0.500 **	0.041	0.604 **	0.039
2009	0.442 *	0.053	0.427 **	0.015
2011	0.508 ***	0.010	0.562 **	0.029
2015	0.658 **	0.032	0.530 **	0.029

Note: *, **, and *** correspond to the statistical significance at 10%, 5%, and 1% level, respectively.

**Table 3 ijerph-16-04545-t003:** Sensitivity analysis with respect to precision parameter τ.

Parameter	Priors 1	Priors 2	Priors 3
	Mean (95%CI)	Mean (95%CI)	Mean (95%CI)
**Exp(b)**			
b1(Males)	0.706 (0.645–0.772)	0.706 (0.645–0.773)	0.704 (0.641–0.770)
b2(Females)	0.986 (0.927–1.048)	0.990 (0.930–1.051)	0.990 (0.930–1.051)
**Exp(β)**	1.161 (0.978–1.386)	1.201 (0.948–1.519)	1.206 (0.953–1.530)
**Exp(S_j_)**			
Males	0.917 (0.855–0.983)	0.916 (0.855–0.982)	0.917 (0.853–0.982)
Females	0.999 (0.966–1.034)	0.999 (0.966–1.033)	0.999 (0.966–1.034)
DIC (pD)	DIC = 1369 (pD = 33.12)	DIC = 1477 (pD = 25.97)	DIC = 1343 (pD = 19.13)

Deviance information criterion (DIC), priors 1: τ_β_~gamma (5.0,5.0 × 10^−4^); priors 2: σ_β_~unif(0,1); priors 3: σ^2^_β_~norm(0.0,100)I(0,). For brevity, this table mainly presents the results of sensitivity for RR.b, RR.β, and RR.S_j_ when *t* = 1 and region i = 2, other results can be obtained upon request.

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
