# Peer review of "Spatio-Temporal Variation of Gender-Specific Hypertension Risk: Evidence from China"

_ijerph, 2019, doi:10.3390/ijerph16224545_

Round 1
Reviewer 1 Report
It was a pleasure reading this manuscript. Only few questions regarding the design and the discussion
The prevalence of hypertension is usually higher among adults. In term of exclusion criteria, it is therefore possible that age 12 to 18 have the lowest prevalence of hypertension. The authors estimated the prevalence of hypertension to be 34.5% in 2015. Is it possible that excluding age 12 to 18 would bring the prevalence much higher? In Line 476-477, the authors state: “the hypertension risk for males in Jiangsu province has started to become salient since 2009. Thus, special attention should be paid to these areas for hypertension prevention and control, especially for males in Jiangsu province”. Please discuss the factual or possible reasons associated with higher prevalence of hypertension among males in Jiangsu province compared to other provinces Section 3 (Results) and sub-section 3.3 (results) have the same title. Please reviewAuthor Response
Dear Reviewer:
I am pleased to resubmit for publication the newly revised version titled “Spatio-Temporal Variation of Gender-Specific Hypertension Risk: Evidence from China” (ID: ijerph-621961) to International Journal of Environmental Research and Public Health. Thanks very much for your kind work and the valuable suggestions during the reviewing process. The manuscript has been revised according to your comments of reviewers’ and we believe our manuscript is much improved as a result.
Response to reviewer#1’s comments:(1) The prevalence of hypertension is usually higher among adults. In term of exclusion criteria, it is therefore possible that age 12 to 18 have the lowest prevalence of hypertension. The authors estimated the prevalence of hypertension to be 34.5% in 2015. Is it possible that excluding age 12 to 18 would bring the prevalence much higher?.
Answer: Thank you for your valuable comments and suggestions. Firstly, I have checked the manuscript and found that the population below age 12 was excluded rather than the population aged 12 to 18. Please see Section 2.1.1 Data Source, page 2, from line 92 to 96:
What’s more, some records in the dataset were excluded according to our eligibility criteria: missing data of age, sex, systolic blood pressure (SBP) or diastolic blood pressure (DBP) (n = 9062); and age below 12 years (n = 16194). Also, …. In the end, 70,374 records of 21,006 individuals aged 12 years and over were retained.
Also, please see the Abstract Section in page 1, from line 15 to line 17:
From the longitudinal data of the China Health and Nutrition Survey (CHNS), 70,374 records of 21,006 individuals aged 12 years and over were selected for this study.
And also, please see the Results Section-3.1 Statistical Descriptions in page 7, from line 269 to line 270:
In total, 70,374 records of 21,006 individuals aged 12 years and over were included in our study, of which 47.5% were males.
Secondly, per your suggestion, the study population was divided into the following four age groups: 12.0-17.99 (Minor), 18.0-44.99 (Young), 45.0-59.99 (Middle-Aged), and >=60 (Old) according to the Report on Chronic Disease Risk Factor surveillance in China (2010) edited by China Center for Disease Control and Prevention. And the hypertension prevalence was 0.84% (Minor)、7.56%(Young)、25.34% (Middle-Aged) and 46.95% (Old), respectively. Therefore, it is true that the prevalence of hypertension is usually higher among adults, especially for the Old Group.
Thirdly, per your suggestion, the age distribution by region was investigated according to the age groups mentioned above. And we found that there was no significant difference among regions, with the Minor Group, Young Group, Middle-Aged Group and Old Group accounted for about 10%、40%、30% and 20% (see Table 1). Therefore, the sample distribution among regions was well balanced, therefore, the prevalence of hypertension was reasonable.
Table 1. Age Distribution by Region (%)
|
|
Minor Group (12.0-17.99) |
Young Group (18.0-44.99) |
Middle-Aged Group (45.0-59.99) |
Old Group (>=60) |
|
Jiangsu |
7.03 |
40.29 |
27.48 |
25.21 |
|
Shandong |
7.36 |
39.33 |
29.55 |
23.76 |
|
Henan |
7.85 |
41.55 |
28.34 |
22.26 |
|
Hubei |
9.44 |
42.40 |
28.32 |
19.84 |
|
Hunan |
9.53 |
42.64 |
28.16 |
19.68 |
|
Guangxi |
9.53 |
41.63 |
25.36 |
23.49 |
|
Guizhou |
11.47 |
40.37 |
25.39 |
22.77 |
(2) In Line 476-477, the authors state: “the hypertension risk for males in Jiangsu province has started to become salient since 2009. Thus, special attention should be paid to these areas for hypertension prevention and control, especially for males in Jiangsu province”. Please discuss the factual or possible reasons associated with higher prevalence of hypertension among males in Jiangsu province compared to other provinces.
Answer: Thank you for your valuable comments and suggestions. Per your suggestion, the possible reasons associated with higher prevalence of hypertension among males in Jiangsu province compared to other provinces were investigated. And we found that overweight and excessive sodium intake may be two important factors contributing the higher prevalence of hypertension. First, the prevalence of overweight for males in Jiangsu province was 20% higher than the average value of the whole population of the sample. Also, the risk of overweight has been identified both in previous studies (see reference [3, 8]) and our study in Discussion Section in page 20, from line 486 to 488: Similar effect of age ….while the effect of overweight on males was more significant, which was consistent with the findings of previous studies [3, 8].
Secondly, the risk of excessive intake of sodium has been identified in previous studies (see reference [51, 52, 53]). In order to better understand the role of excessive sodium intake, reference 53 was added in Discussion Section in page 21, line 497 and in Reference Section in Page 28, line 736-738 as follows:
Xu J, Chen X, Ge Z, et al. Associations of Usual 24-Hour Sodium and Potassium Intakes with Blood Pressure and Risk of Hypertension among Adults in China’s Shandong and Jiangsu Provinces. Kidney Blood Press Research, 2017, 42:188-200. doi: 10.1159/000475486.According to the study of Jianwei Xu et al. (2017) [53], there was more sodium intake for males than females; also, the usually sodium intake was relatively higher for adults in Xinyi & Ganyu of Jiangsu and Fushan & Gaomi of Shandong, especially for the former (for more information, please reference 53 in page 193, Table 2).
Thirdly, though the sodium intake was not incorporated in our model, we found that the hypertension prevalence was relatively higher in spatially adjacent provinces (e.g. Jiangsu, Shandong and Henan). Therefore, combing with the fact that the sodium intake was relatively higher in these areas, we may predict that the excessive sodium intake was associated with the higher prevalence in these regions. Please also see the discussion in the Discussion Section in page 21, from line 495-497:
Furthermore, the relatively higher hypertension risk for people living in spatially adjacent provinces (e.g. Jiangsu, Shandong and Henan) may also be associated with the similar flavor preference in these areas, e.g. their higher salt diet [51, 52].
Moreover, the lack of clear identification between excessive sodium intake and higher hypertension risk in our study has been added as a limitation of our study in Discussion Section as follows (see page 22, from line 537 to 541):
Fourth, the association between the excessive sodium intake and the higher hypertension risk in Jiangsu, Shandong and Henan was not further analyzed due to time constraint. Therefore, future research should take into account more detailed information about alcohol consumption and sodium intake and further identify the potential gender-specific risk factors at regional levels.
(3) Section 3 (Results) and sub-section 3.3 (results) have the same title. Please review.
Answer: Thank you for your valuable comments and suggestions. Per your suggestion, the title of sub-section 3.3 was changed to Results of The Selected Models (see Sub-section 3.3 in page 9, line 307). Thank you!
We thank you for your consideration of this manuscript.
With best regards,
Li Xu
PhD, Division of Statistics
School of Mathematical and Statistics, Guangdong University of Foreign Studies
Zhongda West Road, Xiaoguwei Street, Guangzhou, Guangdong, 510006, China.
Phone: +8613928890096
Email: xuli85@126.com
Reviewer 2 Report
In this study authors aimed to investigate the spatio temporal variation of hypertension prevalence in male and females in China.
The results show that the prevalence of hypertension in China increased from 11.7% to 34.5% from 1991 to 2015, with a higher rate in males than that in females. Hypertension was found mainly clustered in spatially adjacent regions, with a significant high-risk pattern in Eastern and Central China and a low-risk pattern in Western China.
The changes in hypertension prevalence with time and in different regions was was associated with regional covariates, such as age, overweight, alcohol consumption and smoking. Age had similar effect of BP changes in both genders while smoking and alcohol consumption and overweight had different influence of hypertension prevalence changes according to gender, ie the effect of tobacco use on females was more significant than that on males while the effect of overweight on males was more significant,
Authors have analysed a very large number of subjects , and reached interesting findings
this reviewer is not familiar with the quite complex statistical analysis used by authors
The practical implication of the results is clear
No data are reported on the possible effect of antihypertensive treatment use . Could authors give some more information /speculation .
Author Response
Dear Reviewer:
I am pleased to resubmit for publication the newly revised version titled “Spatio-Temporal Variation of Gender-Specific Hypertension Risk: Evidence from China” (ID: ijerph-621961) to International Journal of Environmental Research and Public Health. Thanks very much for your kind work and the valuable suggestions during the reviewing process. The manuscript has been revised according to your comments of reviewers’ and we believe our manuscript is much improved as a result.
Response to reviewer#2’s comments:(1) No data are reported on the possible effect of antihypertensive treatment use. Could authors give some more information /speculation.
Answer: Thank you very much for your comment and valuable suggestion. Per your suggestion, the dataset was further analyzed in order to see the proportion of patients with antihypertensive treatment use and the effect of antihypertensive treatment use.
Firstly, the patients with antihypertensive treatment use was identified from the whole sample. Table 2 showed that about 60% of records were missing regarding the information about anti-hypertension treatment use. This may be associated with the fact that many participants were not aware of their blood pressure condition until they were diagnosed hypertension, especially for the male participants. Also, about 73.4% (4374/5956) of participants with awareness had self-report of taking anti-hypertension medications. An interesting finding is that for participants with awareness of hypertension, the proportion of treatment use was lower in male population than that in female population (70% vs. 76.4%). To conclude, these results tell us that the awareness of hypertension and antihypertensive treatment use was relatively low in China, especially for the former. Though it is not completely comparable, our findings was consistent with previous studies (see reference [52]: section Blood pressure and awareness, treatment and control of hypertension Paragraph 2, page 4).
Table 2. Sample distribution of participants by taking anti-hypertension medications
|
Taking Medications ? |
The whole sample |
Males |
Females |
|
0(No) |
10.9%(1582/14,568) |
11.2%(834/7,468) |
10.5%(748/7,100) |
|
1(Yes) |
30.0%(4,374/14,568) |
26.1%(1946/7,468) |
34.2%(2,428/7,100) |
|
9(Unknown) |
0.1%(7/14,568) |
0.1%(4/7,468) |
0.1%(3/7,100) |
|
.(Missing) |
59.1%(8,605/14,568) |
62.7%(4,684/7,468) |
55.2%(3,921/ 7,100) |
Secondly, the overall effect of antihypertensive treatment use was analyzed by separating the subpopulation with antihypertensive treatment use from the whole sample. To be clear, means SBP of less than 140 mm Hg or DBP of less than 90 mm Hg was defined as control of hypertension among those treated participants. Table 3 showed that about 28% of patients undergoing anti-hypertension medications had their blood pressure controlled. Therefore, the overall effect of taking antihypertensive treatment use was still in low level. This finding was also consistent with previous studies (see reference [52]: section Blood pressure and awareness, treatment and control of hypertension Paragraph 2, page 4).
Table 3. Effects of Taking Anti-Hypertension Medications
|
Had Hypertension Controlled? |
In total |
Males |
Females |
|
1(Yes) |
28.5%(1,245/4,374) |
28.9%(563/1,946) |
28.1%(682/2,428) |
|
0(No) |
71.5%(3,129/4,374) |
71.1%(1,383/1,946) |
71.9%(1,746/2,428) |
Thirdly, the long-term effect of antihypertensive treatment use was also analyzed. As the data were extracted from the China Health and Nutrition Survey (CHNS), an ongoing large-scale survey, the long-term effect of antihypertensive treatment use could also be obtained by selecting the repeated individuals during 1991 to 2015. And we found that 27.5% of patients had their blood pressure controlled for those with anti-hypertension medications during the repeated survey. However, more than 30% of patients had their blood pressure controlled for those who were being surveyed more than six waves and those who were taking antihypertensive medications no more than four waves. This means that the effect of hypertension medications may decrease due to the resistance of antihypertension medications in the long term. Thus, regular monitoring of blood pressure is important for effectively controlling of hypertension.
We thank you for your consideration of this manuscript.
With best regards,
Li Xu
PhD, Division of Statistics
School of Mathematical and Statistics, Guangdong University of Foreign Studies
Zhongda West Road, Xiaoguwei Street, Guangzhou, Guangdong, 510006, China.
Phone: +8613928890096
Email: xuli85@126.com